

# Characterisation of the human uterine microbiome in non-pregnant women through deep sequencing of the V1-2 region of the 16S rRNA gene

Hans Verstraelen[1], Ramiro Vilchez-Vargas[2], Fabian Desimpel[3], Ruy Jauregui[4], Nele Vankeirsbilck[1], Steven Weyers[1], Rita Verhelst[1], Petra De Sutter[1], Dietmar H. Pieper[4] and Tom Van De Wiele[2]

[1] Department of Obstetrics and Gynaecology, Ghent University, Ghent, Belgium
[2] Laboratory of Microbial Ecology and Technology (LabMET), Ghent University, Ghent, Belgium
[3] Faculty of Medicine and Health Sciences, Ghent University, Ghent, Belgium
[4] Microbial Interactions and Processes (MINP) Research Group, Helmholtz Centre for Infection Research, Braunschweig, Germany

Corresponding author
Hans Verstraelen,
hans.verstraelen@ugent.be

## ABSTRACT

**Background.** It is widely assumed that the uterine cavity in non-pregnant women is physiologically sterile, also as a premise to the long-held view that human infants develop in a sterile uterine environment, though likely reflecting under-appraisal of the extent of the human bacterial metacommunity. In an exploratory study, we aimed to investigate the putative presence of a uterine microbiome in a selected series of non-pregnant women through deep sequencing of the V1-2 hypervariable region of the 16S ribosomal RNA (rRNA) gene.

**Methods.** Nineteen women with various reproductive conditions, including subfertility, scheduled for hysteroscopy and not showing uterine anomalies were recruited. Subjects were highly diverse with regard to demographic and medical history and included nulliparous and parous women. Endometrial tissue and mucus harvesting was performed by use of a transcervical device designed to obtain endometrial biopsy, while avoiding cervicovaginal contamination. Bacteria were targeted by use of a barcoded Illumina MiSeq paired-end sequencing method targeting the 16S rRNA gene V1-2 region, yielding an average of 41,194 reads per sample after quality filtering. Taxonomic annotation was pursued by comparison with sequences available through the Ribosomal Database Project and the NCBI database.

**Results.** Out of 183 unique 16S rRNA gene amplicon sequences, 15 phylotypes were present in all samples. In some 90% of the women included, community architecture was fairly similar inasmuch *B. xylanisolvens*, *B. thetaiotaomicron*, *B. fragilis* and an undetermined *Pelomonas* taxon constituted over one third of the endometrial bacterial community. On the singular phylotype level, six women showed predominance of *L. crispatus* or *L. iners* in the presence of the *Bacteroides* core. Two endometrial communities were highly dissimilar, largely lacking the *Bacteroides* core, one dominated by *L. crispatus* and another consisting of a highly diverse community, including *Prevotella* spp., *Atopobium vaginae*, and *Mobiluncus curtisii*.

**Discussion.** Our findings are, albeit not necessarily generalizable, consistent with the presence of a unique microbiota dominated by *Bacteroides* residing on the endometrium of the human non-pregnant uterus. The transcervical sampling approach may be influenced to an unknown extent by endocervical microbiota, which remain uncharacterised, and therefore warrants further validation. Nonetheless, consistent with our understanding of the human microbiome, the uterine microbiota are likely to have a previously unrecognized role in uterine physiology and human reproduction. Further study is therefore warranted to document community ecology and dynamics of the uterine microbiota, as well as the role of the uterine microbiome in health and disease.

## INTRODUCTION

The human body harbours a vast number of bacteria, organised in distinct communities associated with various skin sites, mucosal tract surfaces, and even deeper tissues (*Ribet & Cossart, 2015*), whereby site-specific host-microbe interactions (*Faust et al., 2012*) are collectively found to be essential to many aspects of human physiology (*Dethlefsen, McFall-Ngai & Relman, 2007*; *Cho & Blaser, 2012*). Even body niches widely cited as physiologically devoid of bacteria, such as the lung (*Dickson et al., 2015*) and the urinary bladder (*Brubaker & Wolfe, 2015*; *Whiteside et al., 2015*), have recently been shown to harbour unique microbiota, indicating that conventional compartmentalisation of the human body in sterile and non-sterile body cavities reflects under-appraisal of the extent of the human bacterial metacommunity.

The sterile womb paradigm, coined by French paediatrician Henry Tissier at the turn of the twentieth century, is another enduring dogma also as a premise to the widely held view that human infants develop within a sterile environment (*Funkhouser & Bordenstein, 2013*), though not supported by empirical evidence. During the second half of the 20th century, several researches have challenged the paradigm of the sterility of the uterus through culture of endometrial samples obtained by different approaches, including transcervical sampling with special devices that aim at minimizing the risk of cervicovaginal contamination, perioperative transfundal aspiration, and direct sampling of the endometrial cavity after hysterectomy (*Butler, 1958*; *Bollinger, 1964*; *Mishell et al., 1966*; *Ansbacher, Boyson & Morris, 1967*; *Spore et al., 1970*; *Grossman et al., 1978*; *Pezzlo et al., 1979*; *Sparks et al., 1981*; *Knuppel et al., 1981*; *Heinonen et al., 1985*; *Nelson & Nichols, 1986*; *Eschenbach et al., 1986*; *Teisala, 1987*; *Hemsell et al., 1989*; *Cowling et al., 1992*; *Møller et al., 1995*). Rates of demonstrable bacterial colonisation of the endometrium varied widely in these studies from zero (*Teisala, 1987*) up to 89.0% (*Hemsell et al., 1989*).

With the advent of 16S rRNA gene-based bacterial detection and identification techniques, the sterile womb paradigm has been briefly revisited in recent years. We have

previously shown through fluorescence in situ hybridization (FISH) with 16S rRNA-targeted probes that in pregnant and non-pregnant women with bacterial vaginosis, half of the patients present with a polymicrobial *Gardnerella vaginalis* biofilm that spreads from the vagina into the uterus covering the endometrium (*Swidsinski et al., 2013*). *Mitchell et al. (2015)* recently studied endometrial colonisation by vaginal bacteria, by targeting a series of 12 bacterial taxa through qPCR in hysterectomy patients, including three keystone *Lactobacillus* species and nine bacterial vaginosis indicator species, and found that in 52 out of the 58 women included, at least one of the selected vaginal species was present in the uterine cavity. Hence, as previously hypothesized (*Viniker, 1999*; *Espinoza, Erez & Romero, 2006*), the view of the endometrial cavity as a sterile body compartment may not be longer tenable. In the present study, we aimed to explore the putative presence of a uterine microbiome in a cohort of non-pregnant women with reproductive failure, using a barcoded Illumina paired-end sequencing method targeting the V1-2 hypervariable region of the 16S ribosomal RNA (rRNA) gene.

## METHODS

### Patient recruitment and ethical considerations

Between March to June 2013, consecutive patients with reproductive failure attending our outpatient hysteroscopy facility were invited to participate in the study. Basically, we recruited specifically these patients as they were all reproductive-aged women, including nulliparous and parous women, in whom hysteroscopy was performed immediately following endometrial sampling, which allowed us to exclude any visible uterine anomaly. All study participants gave their oral and written informed consent for endometrial sample collection and subsequent microbiological analysis. All experiments were performed in accordance with relevant guidelines and regulations. Ethical approval was obtained from the Ghent University Hospital Institutional Review Board under reference EC2013/053.

### Patient characteristics

Patients included ($n = 19$) were Belgian or Dutch residents of white Caucasian origin who were referred to the Ghent University Hospital Department of Reproductive Medicine for recurrent implantation failure ($n = 11$), recurrent pregnancy loss ($n = 7$), or both ($n = 1$), and who underwent a hysteroscopic examination as part of the diagnostic work up. Study participants had a median age of 32 years with a range of 25–39 years. Among patients with recurrent implantation failure six were nulligravid, while five had a history of at least one biochemical pregnancy (median 1, range 1–3). Patients with recurrent pregnancy loss had a history of multiple previous pregnancies (median 4, range 3–6). One patient was referred for both recurrent implantation failure and recurrent pregnancy loss and had five early pregnancy losses. Accordingly, apart from having reproductive failure in common, our limited patient series was highly diverse with regard to a number of clinical characteristics including age, gravidity, parity, educational level, and comorbidity. In all patients it was verified that no pregnancy or intra-uterine procedure was documented for

at least six months preceding their inclusion in the study. In none of the patients could uterine anomalies be documented during hysteroscopy.

## Endometrial sampling approach

Endometrial samples in previous studies with the same study goal have been obtained by different approaches, including transcervical sampling with special devices that aim at minimizing the risk of cervicovaginal contamination, perioperative transfundal aspiration, and direct sampling of the endometrial cavity after hysterectomy. As transcervical approaches have been criticized in the past for potential cervicovaginal contamination, we reviewed at the outset of study, the existing literature and carefully considered our options with regard to the different approaches described to obtain endometrial samples, while aiming for reproductive-aged women. Perioperative transfundal needle aspiration approaches are likely to be biased, as such approaches do not readily allow to sample a broad endometrial area with sufficiently deep endometrial tissue harvesting, needle aspiration specimens typically confined to a small volume of endometrial fluid. Approaches where the uterine cavity can be sampled directly clearly are a superior method to the study goal, though also a major constraint to the study of various patient sets, as only in highly selected patients the uterus is removed or opened during surgery. In addition, in our setting, hysterectomy in otherwise healthy, premenopausal women in the absence of oncological conditions, severe comorbidity, or perioperative antibiotic treatment is an uncommon indication, and mostly performed because of dysfunctional uterine bleeding and/or benign uterine wall tumours, potentially altering the intra-uterine environment and affecting uterine bacterial colonisation as previously documented (*Kristiansen et al., 1987*; *Larsson et al., 1990*; *Møller et al., 1995*; *Bhattacharjee et al., 2000*). Accordingly, we did opt for a transcervical approach with the Tao Brush™ IUMC Endometrial Sampler (Cook OB-GYN, Bloomington, Ind., USA), a device that appeared suitable, albeit undoubtedly imperfect, as further specified.

## Endometrial sampling procedure

Patients assumed a classic dorsal lithotomy position for the endometrial sampling procedure and the subsequent hysteroscopy. All procedures were performed by two gynaecologists (NV and SW) with substantial experience in performing intra-uterine procedures and adhered to a strict study protocol. A non-lubricated, sterile, stainless steel Collin speculum was inserted into the vagina to allow for proper visualisation of the ectocervix and the external cervical os in particular. Subsequently, the cervical surface and external os were thoroughly rinsed with an aqueous 0.5% chlorhexidine gluconate solution.

As the endometrial sampling device we used the Tao Brush™ IUMC Endometrial Sampler (Cook OB-GYN, Bloomington, Ind., USA), an FDA Class II device. This particular device has been developed at the Indiana University Medical Center and is primarily intended for the early detection of endometrial carcinoma (*Tao, 1997*). Briefly, the endometrial sampling device is equipped with a brush that is protected by a plastic covering sheath laterally and by a small plastic bead on top to protect the brush on all sites from contamination during passage through the vaginal lumen and endocervical canal.
In the present study, the Tao Brush^TM IUMC Endometrial Sampler was carefully inserted into the vagina thereby avoiding contact with the vulva and the vaginal introitus. During passage through the vagina the device was allowed to make contact with the sterile speculum, but not with the vaginal walls. After insertion of the sheathed brush into the cervical canal the brush was further moved upwards into the uterine cavity, thereby unsheathing the brush, following which the small, flexible brush is rotated five times thereby virtually sampling the entire endometrial surface. The brush is then re-sheathed before it is withdrawn from the uterine cavity. Following the above procedure, the brush was separated in a sterile manner from all other parts of the device. The brush was then placed in a sterile Falcon tube and stored at −80 °C until transport to the laboratory for further processing.

Hence, due to the specific design of the endometrial sampler, and when correctly used, the brush does not make contact at any time during the procedure with the vulva, the vagina, the cervical os, or the endocervical cervical, thereby minimizing the risk of cervicovaginal contamination. In a cytology study of the IUMC Endometrial Sampler, contamination with endocervical cells could be attributed to operators who failed to replace the sheath over the Tao brush before removing it from the uterine cavity, either by inaccessibility of the uterine cavity due to a tight or stenotic cervix (*Maksem, 2000*). It has to be acknowledged however that the performance of the Tao Brush^TM IUMC Endometrial Sampler for the procurement of endometrial samples for microbiological analysis has not been studied before. While we believe that direct contact between the brush and tissue surface before unsheathing in the uterine cavity has been largely prevented due to the stringent study protocol, it cannot be ignored that the protective attributes of the device, the plastic bead on top and the plastic cover sheath did make ample contact with the endocervical canal. The plastic top bead in particular, is inserted into the endometrial cavity and likely introduces endocervical mucus-contained bacteria in the endometrial cavity, though the extent of contamination can be expected to be low relative to the broad and deep endometrial tissue collection, while it is further unknown to which extent endocervical microbiota differ from uterine microbiota.

## DNA extraction and Illumina sequencing

Genomic DNA was extracted essentially as previously described (*Vilchez-Vargas et al., 2013*). Samples were suspended in 1 ml Tris/HCl (100 mM, pH 8.0), supplemented with 100 mM EDTA, 100 mM NaCl, 1% (w/v) polyvinylpyrrolidone and 2% (w/v) sodium dodecyl sulfate, and transferred to a 2 mL Lysing Matrix E tube (Qbiogene, Alexis Biochemicals, Carlsbad, CA) and subjected to mechanical lysis in a FastPrep® −24 Instrument (MP Biomedicals, Santa Ana, California, USA) (40 s, 6.0 m s$^{-1}$) and purified as described (*Vilchez-Vargas et al., 2013*).

The V1-2 region of the 16S rRNA gene was amplified as previously described (*Camarinha-Silva et al., 2014*). However, in a first 20 cycle PCR reaction, the 16S rDNA target was enriched using the well-documented 27F and 338R primers (*Lane, 1991*; *Etchebehere & Tiedje, 2005*) as previously specified (*Chaves-Moreno et al., 2015*).

One µL of this reaction mixture served as template in a second 15 cycle PCR reaction where the forward primer contains a 6 nucleotide (nt) barcode and a 2 nt CA linker and where both primers comprised sequences complementary to the Illumina specific adaptors to the 5′-ends as previously described (*Camarinha-Silva et al., 2014*). One µL of the reaction mixture obtained, served as template in a third 10 cycle PCR reaction using PCR primers designed to integrate the sequence of the specific Illumina multiplexing sequencing primers and index primers. Libraries were prepared by pooling equimolar ratios of amplicons and sequenced on a MiSeq (Illumina, Hayward, CA, USA).

## Data-analysis and reporting

After quality filtering (*Camarinha-Silva et al., 2014*), a total of 782,683 paired-end reads, with an average of 41,194 reads per sample (a minimum of 30,101 reads and a maximum of 62,232 reads) were obtained. All reads were conservatively trimmed to 140 nucleotides and the paired ends subsequently matched yielding 280 nucleotides. Reads were clustered allowing for two mismatches using mothur (*Schloss et al., 2009*). The data-set was then filtered to consider only those phylotypes that were present in at least one sample at a relative abundance >0.1% or that were present in all samples at a relative abundance >0.001%. Accordingly, a total of 676,206 reads were obtained (an average of 35,590 reads per sample with a minimum of 25,756 reads and a maximum of 52,165 reads) and grouped into 183 phylotypes. All samples were randomly re-sampled to equal the smallest read size of 25,756 reads using the phyloseq package (*McMurdie & Holmes, 2013*) from the free software R package for statistical computing and graphics (*R Core Team, 2012*).

Rarefaction curves were generated using the vegan package from the R program (*Oksanen et al., 2007*). All phylotypes were assigned a taxonomic affiliation based on the naive Bayesian classification (RDP classifier) (*Wang et al., 2007*). Phylotypes were then manually analysed against the RDP database using the Seqmatch function (*Cole et al., 2014*) as well as against the NCBI database (*NCBI Resource Coordinators, 2015*) to define the discriminatory power of each sequence read. A species name was assigned to a phylotype when only 16S rRNA gene fragments of previously described isolates of that species showed ≤2 mismatches with the respective representative sequence read. Similarly, a genus name was assigned to a phylotype when only 16S rRNA gene fragments of previously described isolates belonging to that genus and of 16S rRNA gene fragments originating from uncultured representatives of that genus showed ≤2 mismatches. Table S1 gives an overview of the amplicon sequences of the 183 unique phylotypes and their phylogenetic assignment.

Similarities between samples were calculated on a data matrix comprising the percent standardized (untransformed) abundances of all phylotypes using the Bray-Curtis algorithm (*Bray & Curtis, 1957*) with PRIMER (v.6.1.6, PRIMER-E, Plymouth Marine Laboratory). A heat map was generated using the free software R package for statistical computing and graphics (http://www.r-project.org) and the packages gplots (*Warnes et al., 2012*) and RColorBrewer (http://www.ColorBrewer.org), considering only those phylotypes present at an abundance >1% of the total bacterial community in at least one sample.

Since this was an exploratory study on the putative presence of an endometrial microbiome, we aimed for patients with reproductive failure from a mere pragmatic approach, considering these patients had to undergo a hysteroscopy, which also allowed us to confirm the absence of visible uterine anomalies. However, apart from having reproductive failure in common, our limited patient series was highly diverse with regard to a number of clinical characteristics and we therefore refrained from any attempt in correlating clinical and microbiological data.

## RESULTS

### Sampling depth

Rarefaction curves were constructed to estimate whether the sampling depth in each endometrial sample was sufficient to cover the overall bacterial diversity. The curves show that saturation was reached at >15,000 reads per sample (Fig. S1), and hence sufficient for all samples.

### Species diversity

Sequencing of the V1-2 region of the 16S rRNA genes present in the complete endometrial bacterial communities of the 19 subjects with a minimum of 25,756 sequence reads after quality filtering, resulted in a total of 183 bacterial phylotypes, which could be annotated at the phylogenetic levels of Order (93.4% of phylotypes), Family (91.3% of phylotypes), Genus (84.2% of phylotypes) and Species (33.3% of phylotypes). An overview of all 183 bacterial phylotypes along with their relative abundances can be found in Table S2. Out of the 183 phylotypes, 123 phylotypes had a relative abundance of less than 1% of sequence reads per sample in all samples and these phylotypes are therefore provisionally considered as minor components of the uterine microbiome. The highest taxonomical level to which the 60 more abundant phylotypes could be assigned, were Species for 23 phylotypes, Genus for 30 phylotypes, Family for three phylotypes, Order for one phylotype, and Class for three phylotypes, respectively. The endometrial bacterial community structure of the 19 subjects by accounting for the 60 phylotypes with an abundance of at least 1%, is shown as a heat map in Fig. 1.

### Interindividual variability in community structure and core microbiome

An overview of the degree of similarity in bacterial community structure of the 19 endometrial samples is shown in Fig. 2. Twelve out of the 19 bacterial communities (S6, S7, S9, S10, S11, S12, S14, S15, S16, S17, S18, and S19), were quite similar with a mutual similarity of approximately 75% (average Bray-Curtis dissimilarity 24.6%, range 13.2–34.3 %), and hence characterised by the consistent presence of several phylotypes present with comparable abundances. Except for two samples (S15 and S17) also showing a high relative abundance of *L. crispatus*, several phylotypes within the *Bacteroidetes* phylum, were the most abundant taxa in these communities, primarily *Bacteroides xylanisolvens* (Phy1), *Bacteroides thetaiotaomicron* (Phy2), and *Bacteroides fragilis* (Phy7), while *Bacteroides vulgatus* (Phy12) and *Bacteroides ovatus* (Phy20) were although consistently present, less abundant. Several taxa from the *Proteobacteria* phylum were the second most abundant in

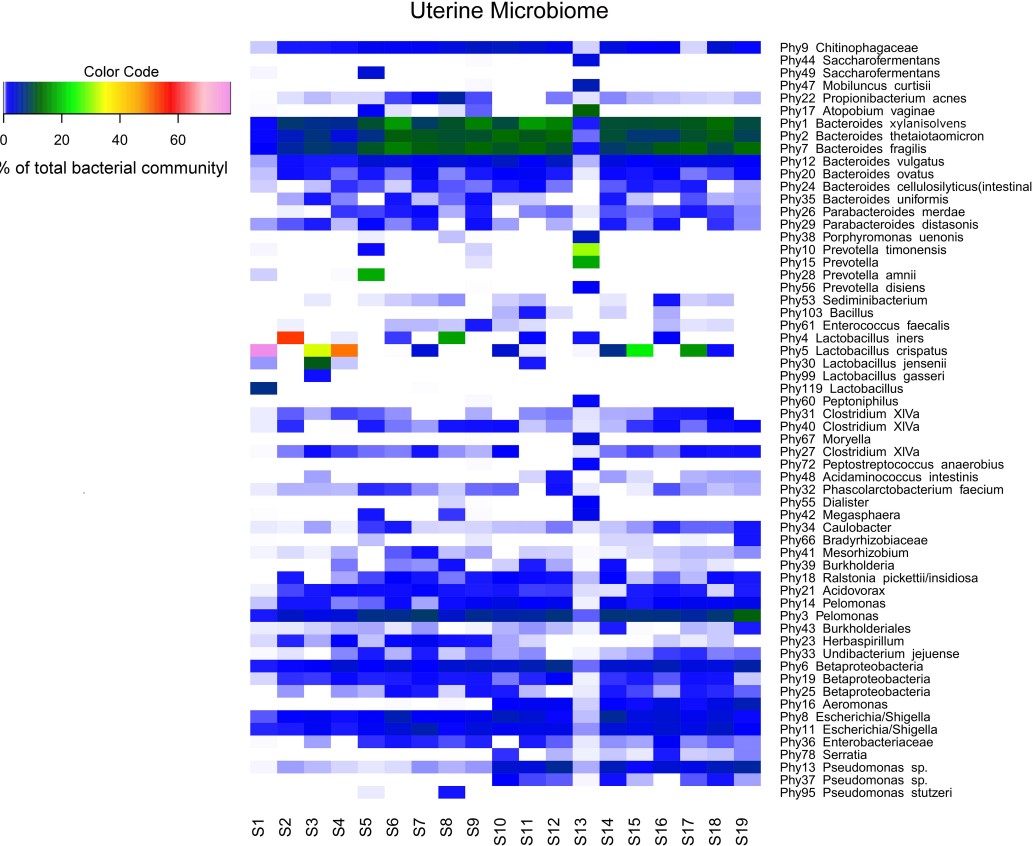

**Figure 1** **Endometrial bacterial community structure in subjects included ($n = 19$).** Endometrial bacterial community structure in each subject, showing those 60 phylotypes which exhibited an abundance of at least 1% in at least one of the samples. The colour code indicates the relative abundance of these phylotypes per sample (see colour code in the upper left corner of the figure).

these communities, including Betaproteobacteria taxa from the *Pelomonas* genus (Phy3 and Phy14) and incompletely assigned Betaproteobacteria (Phy6), and Gammaproteobacteria related to *Escherichia/Shigella* (Phy8 and Phy11). Finally, among the more abundant *Bacteroidetes* were also taxa belonging to *Chitinophagaceae* family (Phy9).

Another 5 out of the 19 bacterial communities (S2, S3, S4, S5, S8) were still similar to the former with regard to the relative abundances of the aforementioned *Bacteroidetes* and *Proteobacteria* taxa, however diverged from the more similar communities due to the co-abundance of typical vaginal taxa: community S8 (Bray-Curtis dissimilarity of 43.6% relative to all other samples) also characterized by *Lactobacillus iners* (Phy4) as an abundant species (18.4% of the overall number of reads), S5 (Bray-Curtis dissimilarity of 45.9% relative to all other samples) showing abundant presence of *Prevotella amnii* (Phy28) (19.1% of the overall number of reads in this sample), S3 and S4 (Bray-Curtis dissimilarity of 50.8 and 53.5% relative to all other samples, respectively), co-dominated by *Lactobacillus crispatus* (Phy5) (35.5% and 52.1% of the overall number of reads in these samples, respectively), and S2 (Bray-Curtis dissimilarity of 60.1%) showing dominance of *Lactobacillus iners* (Phy4) (55.4% of the overall number of reads in this sample).
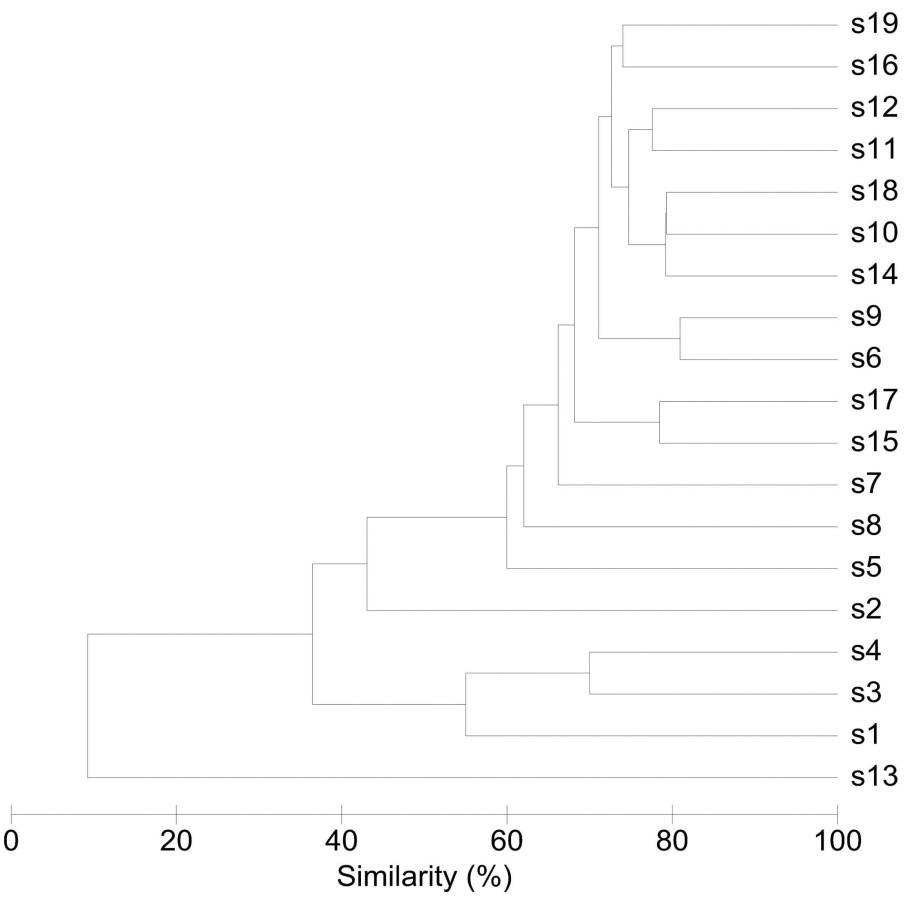

**Figure 2  Similarity between bacterial communities in endometrial samples ($n = 19$).** The dendrogram was constructed by agglomerative hierarchical clustering (group-average) based on a relative abundance matrix of phylotypes. The percentage of similarity between the communities was calculated using the Bray-Curtis similarity algorithm.

Accordingly, while a relatively large bacterial diversity was observed in the intra-uterine environment with 183 phylotypes detected through 16S rRNA V1-2 region sequencing, a defined set of 15 phylotypes with an abundance of at least 1% was observed in all subjects (Table S2 and Fig. 1), suggesting that these phylotypes may constitute the uterine core microbiome. This core microbiome includes the more abundant taxa previously mentioned, *Bacteroides xylanisolvens* (Phy1), *Bacteroides thetaiotaomicron* (Phy2), *Bacteroides fragilis* (Phy7), *Bacteroides vulgatus* (Phy12), *Bacteroides ovatus* (Phy20), *Pelomonas* (Phy3 and Phy14), Betaproteobacteria (Phy6), *Escherichia/Shigella* (Phy8 and Phy11), *Chitinophagaceae* (Phy9), and several taxa that were consistently present, however with lower and variable abundances including *Pseudomonas* (Phy13), *Caulobacter* (Phy34), and *Acidovorax* (Phy21) (Fig. 1). The presumed uterine core microbiome therefore basically consists of three bacterial phyla, in particular *Proteobacteria, Firmicutes* and *Bacteroidetes*, with *Bacteroidetes* dominating the endometrial community in almost 90% of the women included.

Finally, in two subjects, S1 and S13, the endometrial bacterial community largely diverged from all other communities (Bray-Curtis dissimilarity of 79.0 and 90.7%, respectively). In subject S1, the bacterial community was largely dominated by *Lactobacillus crispatus* (Phy5) (79.1% of the overall number of reads in this sample). In subject S13, the endometrial bacterial community was highly different from all other community structures, dominated by *Prevotella* phylotypes (*P. timonensis* (Phy10), *Prevotella* (Phy15), and *P. disiens* (Phy56)), and further characterised by a number of phylotypes uncommon or even unique to the niche in this patient series, such as *Atopobium vaginae* (Phy17), *Porphyromonas uenonis* (Phy38), *Mobiluncus curtisii* (Phy47), *Dialister* (Phy55), *Peptostreptococcus anaerobius* (Phy72), *Peptoniphilus* (Phy60), *Moryella* (Phy67), and *Saccharofermentans* (Phy44).

**Vaginal key species**

A limited number of phylotypes, though not consistently present across all endometrial bacterial communities, were more abundant than the aforementioned core phylotypes in some women. *Lactobacillus crispatus* (Phy5) was present in 12 out of the 19 samples, and the most abundant phylotype compared to the remainder of individual phylotypes in subjects S17, S15, S3, S4, and S1 (17.1%, 25.3%, 35.5%, 52.1%, and 79.1% of the total number of sequence reads respectively). Similarly, *Lactobacillus iners* (Phy4) was present in 7 out of the 19 samples, and the predominant phylotype in two subjects (18.4 and 55.4 % in subjects S8 and S2, respectively). *Prevotella* species in turn, including *Prevotella amnii* (Phy28), *Prevotella timonensis* (Phy10), *Prevotella* (Phy15), and *Prevotella disiens* (Phy56) were predominant in subjects S5 and S13. Noteworthy, *Gardnerella vaginalis* (Phy79) was present in six samples, but always as a minor component (less than 1% of the total sequence reads per sample) of the uterine microbiome. *Atopobium vaginae* (Phy17) was also present in six samples, but only a dominant phylotype in subject S13.

## DISCUSSION

We sought to demystify the longstanding contention that the non-pregnant human uterus is sterile and obtained deep endometrial tissue samples through a transcervical approach from 19 non-pregnant women of reproductive age with reproductive failure and revealed through a barcoded Illumina paired-end sequencing method targeting the V1-2 hypervariable region of the 16S ribosomal RNA (rRNA) gene, that the uterine cavity in this patient series harbours a unique microbiome.

Although a large number of mostly low-abundant phylotypes were identified, the endometrial bacterial community was remarkably similar in a majority of women and characterised by a limited number of particular phylotypes consistently present in a rather similar distribution, presumptively considered the uterine core microbiome. This core assemblage primarily consisted of taxa belonging to the *Bacteroidetes* and *Proteobacteria* phyla, the most abundant core taxa identified as *Bacteroides xylanisolvens*, *Bacteroides thetaiotaomicron*, *Bacteroides fragilis*, and a phylotype belonging to the poorly characterized Betaproteobacteria *Pelomonas* genus. This community assembly pattern was also observed in five women in whom a single *Firmicutes* taxon, *Lactobacillus iners* ($n = 2$) or *Lactobacillus crispatus* ($n = 2$), or *Prevotella amnii* ($n = 1$) were the single most

abundant species. In one of two communities that lacked the typical *Bacteroides* core, *L. crispatus* also largely dominated the endometrial microbiota. Though these *Lactobacillus* phylotypes have recently been shown to be present in the uterus by others (*Mitchell et al., 2015*), such marked abundance of a single species has not been observed with the human microbiota outside the vaginal environment (*Van de Wijgert et al., 2014*), and may therefore be artifactual.

In ~90% of the women included in our study, the genus Bacteroides, primarily *B. xylanisolvens*, *B. thetaiotaomicron*, and *B. fragilis* and to a lesser extent *B. vulgatus* and *B. ovatus* constituted ~30% of the endometrial bacterial community. This is quite similar to what is observed with the human colonic microbiota at the genus level in several studies, though no such consistency has been observed in *Bacteroides* species distribution in the human colon, which is highly variable. Of note is that *B. xylanisolvens*, *B. thetaiotaomicron*, and *B. fragilis* occurred at very similar abundances with the endometrial microbiome in our preliminary study, while in the human colon, *B. fragilis* strains are 10 to 100-fold less abundant compared to other intestinal *Bacteroides* species, constituting less than 1 % of the colonic microbiota. *Bacteroides* species are obligately anaerobic, gram-negative rods and are one of the most numerous bacteria found in the colon of many different animal species (*Bernhard & Field, 2000*; *Thomas et al., 2011*), evidence building that various *Bacteroides* species coevolved with different hosts (*Bernhard & Field, 2000*; *Bäckhed et al., 2005*), also involving substantial genomic changes outside of the 16S rRNA gene (*Atherly & Ziemer, 2014*). Unique features of *Bacteroides* relate to their tremendous, conserved genetic repertoire in utilising a wide variety of carbohydrates, from small sugars to complex polysaccharides, as substrates, as well as their genomic plasticity (*Wexler, 2007*; *Thomas et al., 2011*), likely also explaining their fitness to very different anaerobic environments and hence their abundance as mutualists in the Animalia domain.

*Bacteroides* have been extensively studied with regard to host-bacterial mutualism and are involved in host physiology through a number of mechanisms in the gut, including epithelial cell maturation and maintenance, mucosal barrier reinforcement, and key immunomodulatory functions such as T-cell differentiation (*Wexler, 2007*; *Thomas et al., 2011*; *Furusawa, Obata & Hase, 2015*; *Maier, Anderson & Roy, 2015*), which also enables *Bacteroides* to control their environment by interacting with the host immune system so that it controls other bacteria, such as competing symbionts and pathogens. In this respect, *B. thetaiotaomicron* (*Comstock & Coyne, 2003*; *Zocco et al., 2007*; *Maier, Anderson & Roy, 2015*) and *B. fragilis* (*Wexler, 2007*) have been widely studied and serve as a model to host-bacterial mutualism, further insight gained since the completion of the sequencing projects for *B. thetaiotaomicron* (*Xu et al., 2003*) and *B. fragilis* (*Kuwahara et al., 2004*; *Cerdeño-Tárraga et al., 2005*) and subsequent multi-omics approaches. *B. xylanisolvens* in contrast, which was found in a similar abundance in the endometrial communities as the well-known *B. thetaiotaomicron* and *B. fragilis* taxa, has only recently been described as a novel species in the human gut (*Chassard et al., 2008*), though presumably also widely spread in the Animal domain (*Atherly & Ziemer, 2014*). Further study of the uterine *Bacteroides* is warranted however, as a high diversity of clades at increasing phylogenetic

depth beyond the species and even strain level is expected (*Coyne & Comstock, 2008*; *Atherly & Ziemer, 2014*).

Overall, species diversity and richness of the endometrial communities was significantly higher than in the anatomically closely related vaginal environment and grossly different showing a number of Bacteroidetes and Proteobacteria taxa primarily associated with the gastrointestinal tract, as well as Firmicutes and Actinobacteria taxa commonly found with the vaginal microbiome. The origins of the uterine microbiota within the human bacterial metacommunity remain therefore unclear. Historically, the reluctance to the idea of a non-sterile intra-uterine environment has been attributed to the barrier function of the endocervix in preventing the ascent of bacteria from the vagina, the endocervix having assumed mythic proportions and described as the Colossus of Rhodes of the female genital tract (*Quayle, 2002*). Though the barrier properties of the endocervix have not been fully elucidated, endocervical barrier function is generally attributed to the physical barrier provided by the viscoelastic endocervical mucus (*Linden et al., 2008*) and to unique innate and adaptive mucosal immunity features (*Quayle, 2002*; *Wira et al., 2005*; *Hickey et al., 2011*). Such widely cited theory has been challenged however in many ways. Twenty years ago, *Kunz et al. (1997)* for instance, performed an elegant experiment in which they administered radioactively labelled albumin macrospheres of sperm size at the external cervical os and documented through hysterosalpingoscintigraphy that following their vaginal deposition, albumin macrospheres reached the uterine cavity within minutes. These experiments have been corroborated by others showing rapid spread to the uterine cavity even when radioactively labelled albumin macrospheres were placed at the posterior vaginal fornix at distance of the cervical os (*Zervomanolakis et al., 2007*). By these and related experiments *Kunz et al., (1997)* documented the uterine peristaltic pump function as a fundamental mechanism in human reproduction, the uterus actively harvesting vaginal content. Even the previously considered impregnable endocervical mucus plug that develops in pregnancy, has recently been shown to inhibit though not block the passage of ascending bacteria from the vagina (*Hansen et al., 2014*).

Though the ascent from the vagina therefore appears the most plausible route, it is noteworthy that alternative routes have been suggested with regard to colonisation of the intra-uterine environment, though all related studies have focused on bacterial metacommunity migration related to pregnancy. *Aagaard et al. (2014)* recently reported a comprehensive study of the placental microbiome and pointed at the similarity between the placental and oral microbiota, feeding the concept of a haematogenous oral-placental route (*Mendz, Kaakoush & Quinlivan, 2013*). Of particular interest however, also in view of the findings obtained in the present study, is the concept of entero-placental bacterial trafficking proposed by *Jiménez et al. (2008)*. *Jiménez et al., (2008)* fed pregnant mice orally with a genetically-labelled *Enterococcus faecium* strain and subsequently examined the meconial microbiome of term offspring after sterile caesarean section in a well-designed and controlled study set-up, and detected the labelled strain from meconium of the inoculated animals before the onset of labour, providing strong evidence for entero-placental microbial transmission in mammals.

The implications of the discovery of the uterine microbiome for human health and disease are paramount. Viniker suggested more than a decade ago—even before the term 'microbiome' was coined—that unrecognised endometrial bacterial colonisation might help us to elucidate a number of common gynaecological and obstetric conditions (*Viniker, 1999*). As exemplified by our knowledge on the gut as the most extensively studied human microbiome site, host-microbe interactions are increasing found to be essential to many aspects of human physiology (*Dethlefsen, McFall-Ngai & Relman, 2007*). Accordingly, the upper female genital tract microbiota can reasonably be expected to have a role in uterine physiology and in human reproduction, as recently also suggested by others (*Funkhouser & Bordenstein, 2013*; *Reid et al., 2015*). We have previously documented that subfertile women are considerably more prone to present with dysbiosis of the vaginal microbiome as compared to the background population (*Van Oostrum et al., 2013*). We have further shown that bacterial vaginosis involves the presence of an adherent vaginal polymicrobial biofilm (*Verstraelen & Swidsinski, 2013*) and that this dysbiotic biofilm also adheres to the endometrium in half of the patients presenting with bacterial vaginosis (*Swidsinski et al., 2013*). Hence, albeit the vaginal and uterine microbiomes appear to be quite different bacterial communities residing in completely different physicochemical and immune environments, dysbiosis of the vagina may still predispose to dysbiosis of the uterine microbiome. This would explain for instance the consistent association between dysbiosis of the vaginal microbiome and unfavourable outcomes of human reproduction, such as subfertility (*Van Oostrum et al., 2013*; *Sirota, Zarek & Segars, 2014*), assisted reproductive technology failure (*Van Oostrum et al., 2013*; *Sirota, Zarek & Segars, 2014*) and preterm birth (*Espinoza, Erez & Romero, 2006*; *Mysorekar & Cao, 2014*; *Payne & Bayatibojakhi, 2014*). Interestingly, in our patient series one subject presented with an endometrial bacterial community that was highly different from all other community structures, largely lacking the *Bacteroides* core, while resembling the vaginal microbiome in the setting of bacterial vaginosis, with phylotypes including *Prevotella* spp., *Atopobium vaginae*, *Mobiluncus curtisii*, *Porphyromonas*, *Dialister* spp., *Peptostreptococcus* spp. Our preliminary data do not allow us however to make any firm statements on community states or dysbiosis of the endometrial microbiome.

We do acknowledge that the results of our exploratory study should be taken with considerable caution. Firstly, since we included only a small number of white Caucasian patients with reproductive failure, our microbiome data are not necessarily generalizable to all reproductive-aged women. We explicitly aimed for non-pregnant women without documented uterine anomalies and in whom samples were obtained distant from pregnancy, to avoid any potential influence of gestation on genital tract colonisation. At the same time, it may be recognized that while we designated our patient group under the heading "reproductive failure", study subjects presented with quite different idiopathic reproductive conditions grouped under the umbrella terms "recurrent implantation failure" and "recurrent pregnancy loss". While these reproductive conditions may result from a variety of causes (*Practice Committee of the American Society for Reproductive Medicine, 2012*; *Coughlan et al., 2014*), we specifically aimed to exclude through hysteroscopy all known uterine pathology including fibroids, endometrial polyps,

congenital anomalies and intrauterine adhesions, that have been associated with these conditions (*Practice Committee of the American Society for Reproductive Medicine, 2012*; *Coughlan et al., 2014*). This approach further allowed us to include nulliparous as well as parous women, that were highly diverse with regard to a number of clinical characteristics including age, educational level, and medical history. Hence, while definitely a selected group not representative for the background population of healthy non-pregnant women, it cannot be ignored that a high consistency in endometrial community architecture was observed. Secondly, the most contentious issue in the study of the uterine microbiome since the 1950s has been the debate over cervicovaginal contamination in studies applying transcervical techniques. Alternative approaches have been described, in particular perioperative uterine transfundal needle aspiration and hysterotomy and hysterectomy procedures. Perioperative transfundal needle aspiration is a procedure that might be considered in selected patients undergoing elective abdominal surgery procedures, and allows for the collection of endometrial fluid. Microbiome study of the vagina (*Kim et al., 2009*) and the gut (*Marteau et al., 2001*; *Zoetendal et al., 2002*; *Eckburg et al., 2005*; *Gillevet et al., 2010*) has shown however that luminal bacteria do not adequately reflect the mucosal microbiome, which is also a constraint to gut microbiome studies based on faecal sampling for instance. Approaches were the uterine cavity can be sampled directly during surgery or following hysterectomy clearly provide the most unbiased approach. However, such an approach can only be performed in a limited set of premenopausal women without uterine pathology or major morbidity and hence do prevent further study of the role of the uterine microbiome in various patient sets. Accordingly, we made use of a transcervical approach with a device intended for the procurement of uncontaminated endometrial tissue sampling, which allowed us to sample a broad endometrial area with deep endometrial tissue harvesting. While we maximized efforts as described above to avoid contamination from non-endometrial tissue, we do acknowledge that the sampling procedure likely introduces mucus-contained luminal endocervical bacteria in the endometrial cavity, though the extent of such contamination can be expected to be low relative to the broad and deep endometrial tissue collection, while it is further unknown to which extent endocervical microbiota differ from uterine microbiota. While the inconsistent, yet occasionally pronounced abundance of lactobacilli in particular is therefore potentially of concern in our study, we further obtained a particularly uniform microbiome profile in a majority of study subjects, consistent with other human microbiota showing a few dominant taxa and a long tail of less abundant species. Further validation of our approach is definitely warranted, though it is also clear that further study of the uterine microbiota will rely on transcervical approaches, preferably following further optimisation. Thirdly, we did not control for recent intercourse, though vaginal transfer of seminal bacteria has recently been documented (*Borovkova et al., 2011*; *Mändar et al., 2015*), possibly influencing the uterine microbiota.

We conclude that the present study along with other recent studies are consistent with the presence of distinct microbiota residing in the upper female genital tract including the uterus, fallopian tubes and ovaries (*Pelzer et al., 2011*; *Pelzer et al., 2012*; *Pelzer et al., 2013a*; *Pelzer et al., 2013b*; *Swidsinski et al., 2013*) as part of the human microbiome in women

of childbearing age, and hence that further study is warranted to establish the role of the female genital tract microbiome in women's health and human reproduction.

## ACKNOWLEDGEMENTS

We thank Iris Plumeier and Silke Kahl for their support in the sequencing procedures.

### Funding
The authors received no funding for this work. At the time of the study, Dr. R. Vilchez-Vargas was supported as a postdoctoral fellow by the Belgian Science Policy Office (BELSPO).

### Grant Disclosures
The following grant information was disclosed by the authors:
Belgian Science Policy Office (BELSPO).

### Competing Interests
The authors declare there are no competing interests.

### Author Contributions
- Hans Verstraelen conceived and designed the experiments, analyzed the data, wrote the paper, conceived of the study, obtained ethical approval, coordinated the research.
- Ramiro Vilchez-Vargas analyzed the data, contributed reagents/materials/analysis tools, wrote the paper, prepared figures and/or tables, reviewed drafts of the paper.
- Fabian Desimpel performed the experiments, analyzed the data, contributed reagents/materials/analysis tools, reviewed drafts of the paper.
- Ruy Jauregui contributed reagents/materials/analysis tools, reviewed drafts of the paper.
- Nele Vankeirsbilck reviewed drafts of the paper, recruited patients, obtained samples.
- Steven Weyers conceived and designed the experiments, reviewed drafts of the paper, recruited patients, obtained samples.
- Rita Verhelst and Tom Van De Wiele conceived and designed the experiments, performed the experiments, contributed reagents/materials/analysis tools, reviewed drafts of the paper.
- Petra De Sutter conceived and designed the experiments, reviewed drafts of the paper.
- Dietmar H. Pieper conceived and designed the experiments, performed the experiments, analyzed the data, contributed reagents/materials/analysis tools, wrote the paper, prepared figures and/or tables, reviewed drafts of the paper.

### Human Ethics
The following information was supplied relating to ethical approvals (i.e., approving body and any reference numbers):

Ethical approval was obtained from the Ghent University Hospital Institutional Review Board (Principal Investifator: Hans Verstraelen).

The Ghent University Hospital Institutional Review Board assigned the study reference number EC2013/053.

## Data Availability

All sequences have been listed along with their taxonomic assignment and are provided as supplementary material (Table S1).

Raw data on abundance of unique sequences with their taxonomic assignment are provided as supplementary material (Table S2).

## Supplemental Information

Supplemental information for this article can be found online at http://dx.doi.org/10.7717/peerj.1602#supplemental-information.

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
