# Peer review of "Characterisation of the human uterine microbiome in non-pregnant women through deep sequencing of the V1-2 region of the 16S rRNA gene"

_PeerJ, doi:10.7717/peerj.1602_

## Round 0.1 · original submission · Major Revisions

· Academic Editor

Major Revisions

Dear Hans Verstraelen et.al., your manuscript is unique and interesting yet both reviewers 1 and 2 raised significant questions and comments regarding its methodology and structure that needs to be addressed prior to its acceptance for publication.

Reviewer 1 ·

Basic reporting

In the introduction section, the authors explain the reasons why they want to characterize the uterine microbiomes. However, the introduction is not well structured. Additionally, the authors fail to tell readers the reasons why they use women with reproductive failure rather than healthy non-pregnant women.

Figures are not relevant to the content of the article and appropriately described. For example, Line 268: The result, “Several phylotypes also occurred with low, but highly variable abundances and included bacteria affiliated to Caulobacter (Phy34), Betaproteobacteria (Phy19), Acidovorax (Phy21), Bacteroides ovatus (Phy20), and Pelomonas (Phy14)”, is impossible to be generated based on Figure 2. Furthermore, the Figure 1 legend does not completely describe Figure 1.

Experimental design

Methods are not described with sufficient information to be reproducible by another investigator. For example, the authors simply tell readers that I refer to previous studies to extract genomic DNA and perform Illumina sequencing. However, the authors do not briefly describe how they do them. These two papers, Vilchez-Vargas et al. 2013 and Camarinha-Silva et al., 2014, cannot be freely downloaded. Therefore, it is impossible to reproduce the experiments. Additionally, the authors do not provide the information about how they amplify 16S V1-2 region and what kind of Illumina's platforms they use.

Validity of the findings

No comments

Additional comments

This study characterizes the uterine microbiomes of non-pregnant women with reproductive failure. I recommend that the manuscript will be accepted only if there are major revisions. My comments are that the manuscript is not well-written and needed to be proofread for common surface errors, such as grammar and formatting. I think that a more extensive exploration of the data in the discussion would improve the manuscript. It is currently very descriptive. More emphasis needs to be put on linking results to reproductive failure.

Major comments:
The title is not correct. The authors do not characterize the uterine microbiomes of healthy non-pregnant women.

The methods section is not clear and well structured. The authors do not provide the sample description and tell readers who collects the endometrial samples. The latter part of the DNA extraction and Illumina sequencing section needs to go to the data analysis section. Please explain the reasons why dissimilarity between microbiomes was calculated based on the total number of unique sequences rather than OTUs. Please describe how to generate rarefaction curves and how to generate dendrogram in the heat map. Please deposit your sequences in public website.

In results section, the authors do not provide relative abundance value (percentage) of each bacterium in table. It is hard for readers to follow the contents in the results section.

Minor comments:
Line 66: The sentence “Albeit difficult…” is not clear.
Line 110: Delete “included”
Line 127: What is “os”?
Line 131: “As…” is not a sentence.
Line 157: “et al.” instead of “et al.” Please correct the same error throughout the entire manuscript.
Line 163: The authors should first select an equal number of sequences each sample (based on the sample with the fewest number of sequences) and then obtain unique sequences.
Line 179: It is not clear about how to generate the value in heat maps.
Figure S1: The x and y axis labels of rarefaction curves are not correct.
Table 1: Relative abundance or absolute abundance?
Figure 1: Color key is made based on absolute numbers. However, the authors write that each color represents how many percentage of community in the figure legend. Is the heat map generated based on absolute number or relative number?
Figure 2: Microbiome similarity shown in Figure 2 is different from that in Figure 1.

Reviewer 2 ·

Basic reporting

This is an interesting study aiming to evaluate the presence, and characterize, the uterine microbiome using deep sequencing of the V1-2 hypervariable region of the 16S ribosomal RNA (rRNA) gene. The authors performed this study in 19 non-pregnant women with reproductive failure and without uterine anomalies, and used a transcervical device designed to avoid contamination from the vagina and endocervix in order to sample the endometrial surface. The results of this investigation showed that 15 phylotypes were present in all samples, lead by Bacteroides xylanivorans, Bacteroides thetaiotaomicron, Bacteroides fragilis, and Pelomonas. Moreover, Proteobacteria, Firmicutes and Bacteroidetes were consistently present, and Bacteroidetes dominated the endometrial community in most women. The authors concluded that the uterine microbiome might play an important role in uterine physiology and human reproduction.

The study is very well written and organized, making it easy to read and understand. The manuscript meet the standards of the Journal.

Experimental design

Very well done.

Validity of the findings

My only concern is about the way the sample was obtained. The authors clearly explained how this was done and I believe that most of this is correct. However, to the best of my knowledge, the device used for this purpose has not been studied and validated for microbiologic purposes, other than what is explained in the reference that the authors provided (Tao LC. 1997. Direct intrauterine sampling: the IUMC Endometrial Sampler. Diagnostic cytopathology 17:153-9). I searched the literature for all publications by Tao LC as an author or co-author, and also those including the Tao Brush™ IUMC Endometrial Sampler (searched in different ways), and could not find any study about the use of this device to study the microbiologic status of endometrial samples. If the authors can provide a reference to support the use of this device for the current purpose will be more than welcomed. In the reference mentioned above, the author suggested that this device could be used "... for the procurement of uncontaminated endometrial samples for microbiologic studies from patients with suspected endometritis". The author provided instructions for obtaining uncontaminated endometrial cultures using this device, and mentioned "According to our experience with 16 cases, this device consistently obtains adequate endometrial samples for microbiologic cultures without contamination from the lower genital tract". In my opinion, this reference does not provide strong evidence to support the view that the use of this device avoids contamination from the lower genital tract. This should be demonstrated that before conducting more research, although this is not easy. Maybe the authors should look at patients undergoing abdominal hysterectomy and, at the time of the surgery, perform a puncture of the uterus guided by ultrasound to get the endometrial sample, and then perform the collection transvaginally using this device. It may be complex and cumbersome, but I believe that needs to be done in order to be sure that the use of this device is free of contamination when using in this way.

Additional comments

The results are interesting and add more evidence to that previously published, suggesting that sustaining the idea that the endometrial cavity is sterile is almost impossible. As mentioned by the authors, the characterization of the uterine microbiome may be crucial to understand physiologic and pathologic processes involved in reproduction, such as implantation, recurrent abortion, infertility and maybe also obstetrical complications including preterm delivery, preterm rupture of membranes, or even preeclampsia. Nevertheless, and addressed by the authors, we need to be careful extrapolating these results because the study was performed in patients with reproductive failure.

Reviewer 3 ·

Basic reporting

Characterisation of the human uterine microbiome in non-pregnant women through deep sequencing of the 16S rRNA gene
by Hans Verstraelen et.al.
Peer J 2015:09:6598:0:1
This research is unique research to characterise the uterine Microbiome. Publications describing human microbiome were orientated to vaginal environment considering the vagina as the main source of germs interfering with the woman´s health trough urogenital system. From experience and according to literature the most important issue of all future studies is the proper identification of germs. Considering the vaginal microbiota, many researches in the past were done with simple diagnostic tools with limited spectrum of detection. Using modern identification procedures we could obtain much more information of vaginal microbiota and microbiome.
Considering present study and under circumstances that modern science could not provide proper information on uterine microbiota, the decision to use a 16S targeted Illumina sequencing could elucidate the uterine microbiome up to high level avoiding bias caused by limited diagnostic tools.
I am very pleased that group of infertile woman will be observed in this study. Infertility as such was divided on male or female cause, but the microbiology issue was considered only in limited number (Chlamydia or Ureaplasma spp.). Providing more information on microbiological status of the womb could give us much more information on potential pathology of intrauterine germs.
Comments on protocol:
To avoid the vaginal contamination you used Tao Brush™ for collecting the intra uterine swabs and sterile specula. The Brush is covered by extra sheath, that opens in wombs cavity and closes after obtaining the swab. Have you made any microbiology diagnosis of germs collected only by this sheath going through cervix. This could be a bias considering the potential cervico-vaginal contamination.

Experimental design

Modern and up to date.

Validity of the findings

good presentation

---

## Round 0.2 · accepted · Accept

· Academic Editor

Accept

The manuscript was substantially improved and the reviewe5rs comments were fully adressed

Reviewer 1 ·

Basic reporting

No Comments

Experimental design

No Comments

Validity of the findings

No Comments

Additional comments

I agree that the manuscript has been substantially improved. It is suitable for publication.

Reviewer 3 ·

Basic reporting

Authors answered my questions detailed and update the paper. I am pleased to support this Publication.
No need for further review

Experimental design

"No Comments".

Validity of the findings

"No Comments".

Additional comments

Authors answered my questions detailed and update the paper. I am pleased to support this Publication.
No need for further review